# Phytoplankton Community Structure in Highly-Mineralized Small Gypsum Karst Lake (Russia)

**DOI:** 10.3390/microorganisms10020386

**Published:** 2022-02-07

**Authors:** Alexander Okhapkin, Ekaterina Sharagina, Pavel Kulizin, Natalja Startseva, Ekaterina Vodeneeva

**Affiliations:** 1Department of Botany and Zoology, Institute of Biology and Biomedicine, Lobachevsky State University, 603950 Nizhny Novgorod, Russia; okhapkin@bio.unn.ru (A.O.); ajugareptans@mail.ru (E.S.); KulizinPavel@yandex.ru (P.K.); startseva@bio.unn.ru (N.S.); 2Nizhny Novgorod Branch of the Federal State Budgetary Scientific Institution “Russian Research Institute of Fisheries and Oceanography”, 603116 Nizhny Novgorod, Russia

**Keywords:** phytoplankton, highly-mineralized water, gypsum karst lake, community, structural variables, diversity indices

## Abstract

Gypsum karst lakes are unique water ecosystems characterized by specific habitat conditions for living organisms, including phytoplankton species, as primary producers and mediating biogeochemical cycles in the water bodies. Studies of diversity and structure of phytoplankton communities can be used to identify the specific and typical lake features and plan basin-wide monitoring. The aim of this research was to analyze the structural variables of algocenoses in the small gypsum karstic Lake Klyuchik (Middle Volga basin), atypical for the subzone of mixed coniferous and deciduous forest zone high values of water mineralization (brackish water) and low temperatures. The lake has two water areas, connected by a shallow strait (ecotone zone) and differing from each other in the chemical compositions and physical properties of the water. A total of 133 species of phytoplankton with prevalence percentages of Bacillariophyta (46%), Chlorophyta (24%), and Ochrophyta (11%) were found; α-diversity varied from 4 to 30 specific and intraspecific taxa per sample. According to Spearman’s correlation coefficients, the diversity indices (Shannon, Pielou, Simpson) were mainly determined by the number of dominant species. The uniquely high (up to 130 g/m^3^) biomass of phytoplankton was noted in the ecotone, on the border between the water column and the bottom. The formation of mono- and oligo-dominant nannoplankton diatom communities with a predominance of the rare species *Cyclotella distinguenda* Hustedt was demonstrated there. The roles of flagellate algae and cyanobacteria were found to be less significant.

## 1. Introduction

Karst lakes are widespread types of natural lentic aquatic ecosystems in the landscape world [1,2]. These lakes are mainly small, but relatively deep and often stratified. The hydrochemical facies of these water bodies are typical complexes of predominating solutes, pointing toward definite climatic and, accordingly, geochemical (weathering), soil, and hydrogeological and hydrobiological conditions, under which lake waters acquire their concentrations and compositions [3,4]. Gypsum karst lakes are characterized as sulphate lakes, as they are rich in gypsum or calcium sulphate, and they have elevated conductivity values [5]. These lakes are brackish (dissolve salts up to 2.0 g/L) and often cold water. Some have powerful springs of underground water pressure, specific water balances, amplified water exchanges, high transparency, and azure (ultramarine) water colors [4,6]. On the global scale, surface outcrops of gypsiferous strata appear quite limited [7]. In this regard, such lakes are often considered as endemic [8] or unique [4,5,6,9].

Therefore, these lakes are interesting model systems for the investigation of the different groups of microorganisms. It was found that karst lakes can even have different plankton community compositions and structures compared to other karst lakes of similar geneses and morphometric parameters, even within one catchment area [2,5,6,10,11] Such lakes are interesting biotopes for the formation of special biodiversity; however, its importance has not yet been thoroughly evaluated [5,12].

Phytoplankton is an essential part of water ecosystems, which plays a significant role in food web dynamics, energy flow, and nutrient cycling [13]. Studying the patterns in the species composition and abundance in phytoplankton communities helps to understand, in detail, the complex biogeochemical phenomena in water ecosystems [14]. In addition to environmental factors and the lake’s age, the latitude position has significant effects on the diversity and development of the phytoplankton in lakes [15], including karstic ones.

The phytoplankton of karst lakes located in temperate zones are characterized by co-occurrences of chrysophytes (Chrysophyceae), dinoflagellates (Dinophyceae), and diatoms (Bacillariophyta) as the most diverse and abundant group [16], and, in some lakes, by Cyanobacteria [17] or Chlorophyta [10]. In lakes of the “warm belt”, the dominant role is taken over by Chlorophyta and Cyanobacteria [12,18]. In the spring, high turbulence favors the development and maintenance of diatoms; during the summer, stratification—dinoflagellates, and Cryptophyta, mainly in metalimnion [12,19]. Most of the planktonic algae of karst lakes are cosmopolitan forms. Endemic species were found in the Plitvice lakes [16,20], rare ones—in karst lakes of Greece [12], Romania [21], etc. 

At present, there is little information about phytoplankton community structures assessed by using standard biocenotic metrics [22,23,24,25]. Structural indicators of the phytoplankton community (e.g., species richness, diversity, evenness, dominance, size structure) are rarely described in detail. It reduces the opportunity to determine the main connections that are established in biological communities under certain abiotic conditions [26]. These structural parameters may reflect the influences of a variety of stressors, including climate change and the consequences related with it [27]. They are useful in understanding energy transfer and may be beneficial for more holistic measurements of the health and resilience of lake ecosystems, in general, to multiple stressors. In turn, a detailed study of structural community indicators allows highlighting the features of typicality or the uniqueness of water ecosystems. Such studies are also relevant from the point of view of identifying the biodiversity of aquatic ecosystems, studying the biology and ecology of rare species, and in planning the protection of unique landscapes or habitats [28].

The Middle Volga region in Russia is the zone of the classical manifestation of karst, presented there by various forms [6], including the rarest gypsum karst. Lake Klyuchik represents an example of gypsum karst lakes of sulfate water types. The unique features of the lake are its feeding (by waters of the underground Surin River) [29] and the presence of an ecotone, i.e., a transitional zone between two water areas of the lake where changes in different habitat parameters have been recorded [30]. Due to its unique characteristics the lake has been assigned as a nature sanctuary of regional value [31]. 

Previous studies carried out in Lake Klyuchik focused on taxonomical compositions and the development of phytoplankton [32]. Moreover, we assessed the morphological and ecological parameters of mass species in planktonic algocenoses, *Cyclotella distinguenda* Hustedt [33]. Yet, the structural variables of phytoplankton communities, as well as their spatial and temporal distributions, have not been sufficiently investigated.

The aim of this work was to analyze the various structural parameters of the phytoplankton community, and their spatial and vertical distributions in connection with environmental conditions in the small gypsum karst Lake Klyuchik with unique parameters.

## 2. Materials and Methods

### 2.1. Study Area and Sampling

Karstic Lake Klyuchik (middle Volga basin) is located in an active karst area of Central Russia, Pavlovsky district, Nizhny Novgorod region (56°58′ N, 43°20′ E). Klyuchik is a small lake with a surface area of 11 ha, a maximum depth of 13.5 m, and a mean depth of 3.8 m [34]. The lake is stretched from west to east in an oval shape, with a weakly sinuous coastline and moderately steep coastal slopes. The relief of the lake bottom is uneven, with pits, represented by a sandy-silty substrate [35]. The lake has two water areas, connected by a shallow strait (ecotone zone). Western and eastern parts of the lake differ from each other in the chemical compositions and physical properties of water (transparency, water temperature, oxygen concentration) [30]. 

The source of the lake feeding, the underground spring (the river Surin), is located in its western part and is unloaded in the voklina at a depth of 15 m [29]. The western part of the lake is cold water (temperature varies from 6–10 °C during year) and has low oxygen content (40% saturation) in the surface horizon. The eastern part of Lake Klyuchik is warmer (up to 22–25 °C in summer) and has a high oxygen concentration (more than 115%) [30]. Maximum changes in these parameters are observed in the transitional ecotone zone. 

The western part of the lake is characterized by a uniform vertical distribution of temperature (unstratified), oxygen, and pH throughout the year [30]; it does not freeze in the winter. In the eastern part of the lake, the vertical distributions of environmental parameters are more pronounced, especially during the summer season. The upper warm layer of the epilimnion and the stretched metalimnion were found here. The hypolimnion layer with stable temperature, oxygen, and other parameters were not observed in the lake [30].

The lake is a brackish-water (salinity, ~2 g/dm^3^, electrical conductivity, 1515–1640 μS/cm) [30]. The water color index gradually increases from 40 degree (Platinum-Cobalt scale) in the western part (bluish) of the lake to 62 and 80 degree in the central and eastern (greenish) parts, respectively (Figure 1). The content of phosphates in the water was at the level of permissible values and varied slightly over the lake (0.01–0.05 mg/L). Nitrogen was contained in the water in two forms: nitrate ions 5.2 (mg/L) and ammonium nitrogen (less than 0.01 mg/L) [29]. The sulfate content was high (90–160 mg/L) and could exceed the maximum permissible values up to 7 times.

Lake Klyuchik has an interesting regional importance because of the numerous recreational activities that take place there.

Water samples of phytoplankton (a total of 76 samples) were collected with a Ruttner bathometer and preserved with an iodine–formalin solution in June–August 2017 at five stations (Figure 1), which were chosen by taking into account the limnological and hydrochemical characteristics of the lake. Stations 1 and 2 were set in the deepest western part of the lake, where water has a bluish color. Stations 4 and 5 were located in the central and eastern parts, respectively, which have a greenish water color; station 3 – in the transitional zone. Integrated samples were taken at all stations. Additionally. in July and August, we collected the samples at stations 1, 3, and 5 in increments of 1 m from the surface to the bottom. The water temperature, and pH were measured in situ using a portable Testo sensor, model 206_pH1 (Company Testo SE & Co. KGaA, Lenzkirch, Germany). Transparency (m) was estimated with a Secchi disk (Papanin Institute for Biology of Inland Waters Russian Academy of Sciences, Borok, Russia). 

### 2.2. Data Analysis

In the laboratory the samples were concentrated to 5 mL by combining the settling method and direct filtration [36], and examined under a light microscope (MEIJI Techno, Saitama, Japan) at 1000 magnification. Phytoplankton was analyzed using a 0.01 mL Najott chamber. We estimated the phytoplankton biovolume (mm^3^/L) using geometric shapes closest to the cell shape, taking as a result the mean values of the measurements of 20 to 30 individuals [37,38]. The biomass of each species was calculated by multiplying the number of cells and its biovolume (g/m^3^), total biomass—by summarizing each species biomass [39].

Identification of diatoms was possible due to preparation of permanent slides using Naphrax resin (Brunel Microscopes Ltd, Chippenham Wiltshire, United Kingdom) [40]. Centric diatoms were analyzed with the help of a JSM-25S scanning electron microscope (JEOL Ltd., Tokyo, Japan) [33]. Phytoplankton species were identified based on morphology. Nowadays, the optical method of phytoplankton analysis continues to be the principal approach in the ecological monitoring of the water quality, despite some limitations [28,41]. The list guides used for species identification were performed in previous studies [42]. The current names of taxa were also checked using the AlgaeBase website (https://www.algaebase.org/, accessed on 29 July 2021) [43].

The authors analyzed such parameters of the phytoplankton community (= coenocytic parameters) as follows: abundance (N), 10^6^ cells/L, biomass (B), g/m^3^, species richness (α diversity, Sp—number of species in one sample). Phytoplankton alpha diversity indices were evaluated using the Shannon–Weaver diversity index for abundance (HN, bit/Ex), and biomass (HB, bit/g) [44,45].
(1)$$H=-\sum_{i=1}^{S}P_{i}\cdot log_{2}\;P_{i}$$

Pielou evenness index (EN and EB)
(2)$$E=\frac{H}{H_{max}},$$
where *P_i_* is the relative abundance or biomass of the *i*-th species, *S* is the total number of species in the sample, *H_max_* is the maximum Shannon–Weaver index for a given number of a species.

Simpson dominance index (DN and DB)
(3)$$D=\frac{n_{i}\left(n-1\right)}{N\left(N-1\right)}$$
where *n_i_* is the abundance or biomass of the *i*-th species; *N* is the total abundance or biomass of phytoplankton in the sample

Water saprobity was evaluated by Pantle–Buck indices, which were calculated according to abundance (SN) and biomass (SB) [46].
(4)$$S=\frac{\Sigma S\cdot h}{\Sigma h}$$
where *S* is the indicator significance of the saprobic species-indicator, *h* is the abundance or biomass indicator.

Size structures of phytoplankton communities were estimated two ways—as arithmetic (by geometric shapes) volume (V_am_) of the algae cell in a sample and a coenocytic volume estimated as B/N (Vc, µm^3^). Moreover, we considered the share of a small cell fraction (<20 µm, nannoplankton, according to: [47]) in the total phytoplankton abundance (%N) and biomass (%B). In addition, the dynamics of flagellar forms of phytoplankton were analyzed. They were evaluated by the share of the monad forms in total abundance (%Nflagel) and biomass (%Bflagel) of algae.

The authors defined the dynamics of the number of dominant and co-dominant species, which were extracted by abundance (SDN and SMN, respectively) and biomass (SDB, SMB). The dominant species included species with an abundance or a biomass more than 10% of the total value and the co-dominant species included species with an abundance (biomass) of 5–10% [48]. 

### 2.3. Statistical Analysis

As data do not have normal distribution, the non-parametrical [49] Mann–Whitney criterion (U-criterion) was used to compare the variables; and the Spearman correlation (R_s_) to estimate the relationship among the parameters of phytoplankton communities. Statistical processing was conducted using the Statistica 8.0 software package (Statsoft TIBCO, Palo Alto, CA, USA). The authors discussed reliable connections of parameters at the significance level of *p* ≤ 0.05.

## 3. Results

### 3.1. Environmental Conditions 

Table 1 shows the values of the environmental variables recorded at the sampling stations. The depths of sampling stations ranged from 2 m in the transition zone (St. 3) to 10.7 m in the voklina (St. 1), in the area of the main groundwater supply, and from 7.0 (St. 4) to 9.4 (St. 5) meters at deep-water stations in the eastern part of the lake. The transparency of lake waters in the zone of maximum depths decreased twice from station 1 to station 5. In the western part, the temperature varied from 7.5 to 8.0 °C in June to 10.5 °C in August. 

The eastern part of the lake was warmer; the temperature changed there from 11.3 to 13.2 °C in June to 15.8–17.3 °C in August. During the study period, the pH value ranged between 7.0 and 7.6, with an implicit tendency of increasing from June to August, and a clearly manifested increase from the western part of the lake to the eastern one. In the same direction, the color of the lake’s waters changed from sky-blue to greenish–blue.

### 3.2. Phytoplankton Community Composition 

The phytoplankton community studied was composed of 133 species, including Bacillariophyta 46%, Chlorophyta 24%; Ochrophyta 11%; Cyanobacteria 8%; and Charophyta, Cryptophyta, Euglenozoa, and Myzozoa, less than 3% each. The number of phytoplankton species per sample varied from 4 to 30 taxa (specific and intraspecific) at different stations. We recorded 13–20 taxa in the integrated samples and 4–15 for particular depths in the western part of the lake. In the ecotone zone (St. 3) the number of taxa increased up to 26–30 and 10–29, respectively.

### 3.3. The Spatial Distribution of Phytoplankton Abundance, Biomass, Diversity Indices, Size Structure

Table 2 shows the average (M ± m) values of the structural (coenocytic) parameters of the phytoplankton community in different stations (spatial distribution). The abundance and biomass of phytoplankton fluctuated significantly during summer period.

The lowest average abundance (0.31 million cells/L) and biomass (0.94 g/m^3^) of phytoplankton was recorded at station 1 (voklina). According to the diversity indices in this part of the lake the oligo-dominant algocenoses with medium values of species richness, diversity, evenness, and dominance developed. The phytoplankton community was mainly formed by a small-cell species with an insignificant proportion of flagellar forms (%B flagel—0.4; %N flagel—0.6) among them. 

The northwestern shallower water area of the lake (St. 2) was characterized by a ten-fold increase in the degree of phytoplankton development, with an unreliably pronounced tendency to increase the temperature and pH of the water. The development of 2–4 dominant algae species was accompanied by a decrease in the diversity and evenness of communities. At the same time, the proportion of small-cell fraction and flagellar forms increased (especially in terms of biomass %Bflagel—6.1). At this station, the arithmetic mean volume (V_am_—12.5) of algal cells increased (almost three-fold), while the coenocytic volume (Vc) did not change. This may indicate the appearance in the phytoplankton communities of a few large forms of algae, which did not yet occupy a dominant position.

The development of phytoplankton in the eastern water area of the lake (Sts. 4, 5) did not significantly differ from the station 2, but it was higher, especially biomass (the statistical comparison from the U-criterion showed significant differences (*p* < 0.05)) than in the zone of the main groundwater supply (St. 1). In these sites of the lake, the distribution of the phytoplankton species, in terms of abundance and biomass, was more even. Accordingly, the Shannon indices (especially at St. 4) and the Pielou evenness indices were higher here, and the dominance was less pronounced. In the eastern water area of the lake, the role of large-celled algae species became more significant, whereas the proportions of low-sized components in phytoplankton communities decreased. In these stations, some representatives had sizes of more than 64 microns (net plankton, according to [48]). In addition, the composition of dominant forms became richer, and there was a clear tendency to increase the role of flagellar algae (especially on station 5). 

In the ecotone zone (St. 3), the development of phytoplankton was the most productive. Due to the diatoms “blooming”, monodominant (DN = 0.71, DB = 0.64) communities formed in this part of the lake (Table 2), in which the only one species of centric diatoms—*Cyclotella distinguenda*—dominated. Most of this diatom species had a cell diameter of less than 20 microns, so the share of small cell algae in the abundance (91.6%) and in the biomass (82.8%) in this area of the lake was the maximum.

### 3.4. The Vertical Distribution of Phytoplankton Abundance, Biomass, Dominant Species, Diversity Indices, Size Structure 

Heterogeneity in the distribution of structural parameters of the phytoplankton community was also noted along the depth (Figure 2, Figure 3, Figure 4 and Figure 5).

In the deepest part of the western area of the lake (station 1), two layers with noticeable phytoplankton richness were distinguished (0–1 and 8–9 m in July, 0–2 and 7–9 m in August) (Figure 1 and Figure 2). In the first one (surface layer), the highest values of species richness (32 species), Shannon indices (HN = 3.52, HB = 3.36), and evenness index (EN = 0.70, EB = 0.67) were established in July. In this month, the phytoplankton abundance and biomass were typical for mesotrophic lakes (0.45 million cells/L, 1.37 g/m^3^). Algocenoses were formed with a predominance of *Cyclotella distinguenda* and *Melosira varians* Ag., accompanied by large-celled benthic diatom forms (species of the genera *Pinnularia*, *Caloneis*, *Amphora*, *Navicula radiosa* Kütz., etc.). At the depths of 5–7 m, the development of phytoplankton significantly decreased, increasing again in the bottom layer. Near the bottom, there were the phytoplankton communities with the predominance of large-cell forms (especially in the formation of biomass) and the participation of *Melosira varians*, *Cyclotella distinguenda* and *Meridion circulare* (Grev.) Ag. A few flagellar algae (genera *Chrysococcus*, *Dinobryon*, *Euglena*, *Vacuolaria*) were found at depths from 0.5 to 4.0 m.

In August, at station 1, the vertical distribution of structural parameters of phytoplankton was the same (Figure 2). In the surface water layer, the development of algae was insignificant (the abundance reached 0.1–0.2 million cells/L, the biomass—0.1–0.26 g/m^3^). Near the bottom (7–9 m), the abundant increased 1.5 times, and the biomass—6 times. In the intermediate layer (3.0–6.0 m), the decrease in species richness (up to 4–6 species in one sample) and diversity indices were found. In comparison with surface layers, the participation of the small-celled fraction of plankton in the formation of its abundance became more noticeable (by 1.5–2.2 times), and its contribution to the phytoplankton biomass was maximal (up to 79%). From a depth of 6 m to the bottom, the proportion of large-cell phytoplankton species increased because of the benthic species of the diatoms presence in the sample (species from genera *Nitzschia*, *Navicula*, *Cymbella, Gomphonema*, *Pinnularia*) (Figure 3 and Figure 4). Thus, in the zone of the greatest depths of the western part of the lake, two maxima of phytoplankton development were formed at the surface, and near the bottom in July and at the bottom in August.

At station 5 (in the eastern part of the lake), the maximum phytoplankton development was clearly manifested near the bottom (Figure 2) with predominant large benthic species of diatoms (genera *Pinnularia*, *Amphora*, *Nitzschia*, *Cymbella*). In July, their relative biomass and abundance were more than 60% and 10%, respectively. In contrast to benthic communities, communities on the surface of this station were formed by small-cell species of phytoplankton, such as *Cyclotella distinguenda*, *Hadmannia comta* (Ehr.) Kociolek and Khursevich, *Monoraphidium minutum* (Nägeli) Komárk.-Legn., and *Dictyosphaerium subsolitarium* V. Goor with a significant relative abundance. The phytoplankton of this part of the lake was enriched with flagellate algae, which were found in each layer (22.5 ± 3.2% and 75.9 ± 8.8% of the total abundance and biomass, respectively, for the entire water column). Their relative biomass was the highest at the surface and gradually decreased at the bottom. Among the monad fraction, both a few large-celled species (*Peridinium cinctum* Pénard. and *Ceratium hirundinella* (O. Müll.) Dujard.) and numerous small-celled ones (*Komma caudata* (Geitler) D.R.A. Hill., *Kephyrion* gen. spp.) were distinguished. Dominant species *Peridinium cinctum* developed in a hole water column (from the surface to the bottom), forming for 61–89% of the phytoplankton biomass in the surface and intermediate layers; near the bottom—only 14%. Algocenoses that formed in the bottom layer (depth 7 m) were characterized by high Shannon indices (HN = 3.43; HB = 3.25) and Pielou evenness indices (EN = 0.72; EB = 0.68) of species and weak dominance in abundance (DN = 0.15) and biomass (DB = 0.16) of phytoplankton (Figure 2 and Figure 4).

In August, the vertical distribution of phytoplankton structural parameters at this station was similar to that of July. In the surface layer, the biomass and abundance of phytoplankton corresponded to the oligotrophic state, in a depth of 6–7 m—to the mesotrophic level (9.2 million cells/L – 4.53 g/m^3^) (Figure 3). Increasing the depth, there was a tendency in enlarging of α-diversity, although the number of dominant species in abundance in the bottom layers turned were lower than in the rest of the water column. The phytoplankton community was mainly formed by small-celled species. Their proportion in the total abundance was maximal between layers of 5 to 7 m, while the contribution in the biomass sharply decreased near the bottom (Figure 5). At the deepest layer upon contact with the bottom surface, the benthic species of algae (mainly large-cell pennate diatoms—*Navicula radiosa*, *Amphora ovalis* (Kütz.), *Pinnularia* sp.)—enriched the plankton communities and contributed the increase of the arithmetic volume of cells (V_am_ = 16640) (Figure 3). The relative abundance of flagellate algae increased more than twice in comparison with July, but their proportion in the total biomass remained the same. Dominant in abundance were *Komma caudata*, species of the genera *Cryptomonas*, *Chrysococcus*, *Dinobryon,* and other unidentified small-cell chrysomonads. These species vegetated (up to 8.92 million cells/L) at a depth of 4–5 m to the bottom forming more than 90% of the total abundance on the 6-m layer. Most notable in the creation of biomass were large-celled *Ceratium hirundinella* (up to 1.78 g/m^3^) and *Peridinium cinctum* (up to 2.0 g/m^3^), they prevailed in the bottom layer of water, and *Vacuolaria virescens* Cienk. (up to 0.7 g/m^3^) at a depth of 4.0 m.

In the shallowest, station 3, in the ecotone zone during the summer, the highest integrated abundance and biomass (97.8 g/m^3^) of phytoplankton were observed in August. The vertical stratifications of phytoplankton abundance and biomass were the most pronounced there. Phytoplankton was sharply concentrated in the bottom layer of water (the abundance of this group of living organisms was 32 and 49 times higher than in the surface in July and August, respectively, and the biomass was 21 and 36 times higher; the maximum value of biomass was 130 g/m^3^) (Figure 2 and Figure 4). In this part of the lake, oligo- and even monodominant phytoplankton communities, poor in species, were formed with an absolute prevalence of *Cyclotella distinguenda* near the bottom. Alfa-diversity indices (especially in terms of the abundance of species) had higher values in the surface layers and reached a minimum (Shannon indices: HN = 0.11; HB = 0.34 and Pielou evenness indices: EN = 0.04; EB = 0.11) at the bottom during the period of diatoms “blooming”. In the vertical distribution, there was a reduction of the proportion of flagellar algae from the surface (25.2–37.9% of abundance, 73.9–87.1% of the biomass) to the bottom (0.11 and 0 accordingly), and opposite, gradually increasing the share of small-celled plankton from the surface to the bottom (by 2–5 times in terms of biomass), from the beginning of the summer period to its end (Figure 3 and Figure 5).

### 3.5. Statistical Analysis 

Phytoplankton abundance and biomass in Lake Klyuchik were positive correlated with each other (Rs = 0.89). The abundance tended to increase in more saprobic and trophic waters (the correlation coefficients between N and SB was 0.63). At the same time, no significant relations with the structural variables of phytoplankton communities were revealed, except for the abundance and its evenness index (EN), which had a negative relation (Rs = −0.54). The abundance of phytoplankton positively correlated (Rs = 0.52) with the proportion of the small-cell species (%N) in its formation and negatively correlated with the proportion of the dominant species, in terms of biomass (Rs = −0.59 and −0.69, the relationship of this parameter with the abundance and biomass, respectively). 

The correlation analysis showed that the Shannon–Weaver indices (HN, HB) positively correlated with the Pielou evenness indices, and they had a significant negative correlation with the Simpson dominant indices (Table 3). 

The Shannon indices had a negative relation with the proportion of the small-celled species in the abundance and biomass, saprobity, and a weakly positive correlation, with a relative abundance of flagellar forms and a diversity of dominant and mass species (Table 4). A similar tendency was recorded with Pielou, which had a positive correlation with a majority of structural variables. The Simpson dominant indices had the opposite trend (Table 4).

No significant relationship was found between species richness and other structural variables of phytoplankton communities. The number of dominant species by biomass was negatively related to the abundance (Rs = −0.69) and biomass (Rs = −0.59) of phytoplankton and positively to the coenocytic volume (Rs = −0.57). The number of dominant and subdominant species in abundance had negative correlations with the share of the small-cell species in the abundance (Rs = −0.51 and −0.85 respectively) and biomass (Rs = −0.51 and −0.53) of phytoplankton, i.e., with its increase, the composition of coenose-forming components with a size of less than 20 microns became poorer. Positive correlations were also noted with the share of flagellates, in abundance (Rs = 0.53–0.86). 

The number of dominant species was not determined by the species richness (*p* ≥ 0.05). Usually, it was not notable (2–4 species in a sample, rarely more – up to 7, station 4), as well the total number of species (average for the lake 15 ± 1, maximum 32–35).

There was a positive correlation (Rs = 0.65) between the arithmetic volume of algal cells (V_am_) and coenocytic volume (Vc) in phytoplankton communities of Lake Klyuchik; the connection between these parameters and the abundance and biomass was negative, but unreliable (*p* ≥ 0.05). The low positive correlation of the small-cell species proportion in abundance with the biomass of phytoplankton (Rs = 0.52) and the significant contribution in those parameters (over 40–70%) (Table 4) reflected the predominance of the nannoplankton component in phytoplankton communities. 

Flagellate algae were poorly represented in the phytoplankton communities of the lake. Their share on average for all stations was 4.3–4.5% of the abundance or biomass; at some stations (St. 5) it increased to 11–14% (Table 2). The presence of monadic algae to a certain depth was also unclear. At St. 1, they concentrated at a depth of 3 m (August) and were the only group in that layer, or their noticeable development (up to 87% of the total biomass) was noted in the surface layer (St. 3—ecotone zone); sometimes they created a maximum (4.53 g/m^3^—up to 97% of the total biomass) in the bottom layer (St. 5) (Figure 5). The proportion of flagellate algae in the abundance showed a positive correlation with the arithmetic volume of the algal cell (Rs = 0.71), since the large-sized representatives (*Ceratium*, *Peridinium*, *Gonyostomum*, *Vacuolaria*) or colonial (*Dinobryon*) species of algae were mainly observed there.

A review of relations of phytoplankton structural parameters with environmental parameters demonstrated a significant role of the depth. The depth of the lake had a positive correlation with the number of dominant species (Rs = 0.62–0.66), diversity Shannon–Weaver indices (Rs = 0.76 for HN and Rs = 0.61 for HB), and Pielou evenness (Rs = 0.82 for EN and Rs = 0.61 for EB), and negative with Simpson dominant indices (Rs = −0.68 for DN and Rs = −0.61 for DB). A significant negative correlation was also noted between the depth and water saprobity index by biomass s (Rs = −0.61). 

The value of water transparency had a negative correlation with the abundance (Rs = −0.97) and biomass (Rs = −0.90) of phytoplankton, the water saprobity index by biomass (Rs = −0.77), and the proportion of small-celled algae that played a role in the formation biomass (Rs = −0.80). Transparency correlated positively with diversity index based on abundance (Rs = 0.62), evenness (Rs = 0.59), and the number of dominant species (Rs = 0.73). The temperature factor had positive correlations with the volume of algae cells (Rs = 0.53) and the proportion of the flagellar species in the forming of abundance (Rs = 0.51) only. The value of pH showed a positive relationship only with the proportion of flagellate algae, in terms of abundance (Rs = 0.57).

## 4. Discussion

Lake Klyuchik is a small gypsum karst lake with calcium sulphate water. Like most gypsum reservoirs [4,5,6], it has brackish water (dissolved salts up to 2.0 g/L), high transparency, and an azure (ultramarine) water color. The lake also has some specific characteristics. Firstly, it has an ecotone zone that connects two parts of the lake differing from each other in the chemical composition and physical properties of the water. Secondly, in the western water area of the lake, there is an underground spring, the flow rate of which varies from 1.79 m^3^/s to 4.21 m^3^/s [34]. According to the flow-based classification [50], it belongs to the second-magnitude group of springs; from this point of view, lake Klyuchik can be considered a unique hydrological object. The presence of an underground spring ensures a constant low temperature in the western part of the lake (it does not freeze during the winter) throughout the year, and a lack of temperature and oxygen stratification [30]. With this combination of factors, Lake Klyuchik turned out to be similar to Lake Goluboe (Samara Region, Russia) [17] and Lake Ochiul Beiului (Romania) [2].

The water transparency in Lake Klyuchik was in the range of 3 to 8 m and was peculiar to oligotrophic–mesotrophic types of water bodies [51]. The values of transparency were less in the ecotone zone due to the slight depth at this part of the lake (up to 2 m). The coefficient of relative transparency, estimated as the ratio of the average transparency to the average depth, was 1.1. It is typical to the class of optically deep water bodies, in which the water transparency is 1–2 times greater than the average depth [51], favoring the development of phytoplankton at considerable depths. Lake Klyuchik can be classified as neutral or oligo-alkaline, according to the pH value [51]. 

We regard this lake as a model of an aquatic system, where the effect of the environment "severity" is clearly expressed. According to Bigon et al. [52], the environment "severity" means the predominance among environmental conditions of one or few limiting factors (acidification, pollution, thermification, etc.), which are responsible for influencing the structural variables of the biotic communities and suppressing the effects of other factors. In Lake Klyuchik, two main factors (temperature and mineralization, which have great influence on the structure of algocenoses) are combined in an unusual way, not typical for the lakes of the temperate forest zone. Although the amount of solar energy supplied per unit of the lake surface area is typical for the forest zone, the thermal regime of the lake is more similar for the northern lakes of the coniferous–deciduous forests. Lake Klyuchik, consistent with the thermal regime, is similar to water bodies of the tundra and forest–tundra, for example, the subarctic lakes of Fennoscandia [53]. On the other hand, the values of mineralization are more typical for southern lakes of the steppe zone [6,10,17]. 

The continual inflow of underground cold, blue-colored waters, increases the albedo of the lake in its western part. It contributes to a significant reflection of light energy from the surface, weak water heating (average summer water temperature was 8.8 ± 0.5 °C, seasonal temperature variations were 2.5–3.0 °C), and the formation of stable low-temperature conditions in the summer season. The eastern part of the lake is more productive because its waters are warmer in the summer (average temperature was 14.7 ± 0.9 °C, seasonal temperature variations were 4.1–4.5 °C). The transparency of the waters is about two times lower than in the western part, the waters are greenish–blue in color because of the development of autotrophic plankton. With the same amount of incoming sun energy, its absorption and, accordingly, water heating, are more efficient in the western part of the lake. It contributes to the compensation of the low temperature background and stimulates the development of thermophilic algae species. 

The floristic list of algae in Lake Klyuchik was formed by nine taxonomical groups where diatoms significantly prevailed in terms of the number of species (more than 40%). The proportions of green algae and cyanoprokaryotes were lower, while in the majority of water bodies in the temperate zone, they usually predominated [10,54,55]. The same proportion of large taxa (divisions) of algae were recorded in Lake Goluboe, Russia [17], and in Lake Ochiul Beiului, Romania [2], with similar habitat conditions. The species composition of phytoplankton of Lake Klyuchik was represented mainly by benthic, littoral, and truly planktonic forms with dominance of the cosmopolitan species (87.1%) and a low proportion of boreal ones (12.9%) [32]. In this regard, this water bodies are similar to most of the lakes [15]. 

Species richness is one of the most important parameters of the phytoplankton community [52,56,57]. The specific combination of environmental factors in Lake Klyuchik was weakly beneficial for the formation of species-rich phytoplankton communities there. The alpha diversity of phytoplankton (18–20 species per sample in summer season) was significantly lower than in other waters bodies (e.g., in the lakes of the Pustynsky Reserve—up to 40–50 species per sample) of the same karst zone of the Volga basin [54]. Similarly, low values of α-diversity were noted for highly-humified and acidic water bodies [58,59,60]; in this term, it can be considered a common answer, in regard to biota communities and stress conditions. 

Analysis of the spatial and vertical distribution of the main parameters of the phytoplankton structure of Lake Klyuchik and their correlations revealed some patterns. The distributions of the taxonomic composition, quantitative development, and diversity indices of phytoplankton were characterized by spatial heterogeneity. The abundance and the biomass varied significantly during the summer period and generally characterized Lake Klyuchik as a eutrophic lake [51] with the average biomass being more than 5.0 g/m^3^, except for the area of voklina (St. 1). At this station, the phytoplankton abundance and biomass were not high, peculiar to oligotrophic or the weakly mesotrophic state. In terms of temporary changes, the maximum biomass values at stations 2 and 5 were recorded in July, and at stations 3 and 4—in mid-August. At the deepest, station 1, biomass values were similar during the summer. Successions of the dominant species are presented in previous studies in more detail [32].

The dominant groups of phytoplankton were similar to those noted for other karst lakes of temperate zones, including gypsum ones [2,10,16,21]. The phytoplankton of Lake Klyuchik is characterized by co-occurrences of chrysophytes (Chrysophyceae—up to 20% of the total abundance), dinoflagellates (Dinophyceae—up to 40% of the total biomass), and diatoms (Bacillariophyta) in the warm water part of the lake, and practically complete (50–100%) dominance of diatoms in the cold water part, where they often develop near the bottom. In high transparency conditions (up to 8.5 m), and a lack of light limitation, photosynthesis was possible throughout the entire water column, including the area near the benthal zone. This area became available for the normal functioning of planktonic or benthic algae, including representatives of the "shadow" flora, mainly from the diatoms. Avoidance by diatoms of well-lit surface layers of highly transparent water bodies (both marine and fresh) is a well-known fact in the ecology of phytoplankton and diatoms [47] (the phenomenon of a deepened maximum of photosynthesis).

The proportion of dominance groups and individual development values of dominant species were rather specific. In this regard, the ecotone zone of the lake is of particular interest, where the most noticeable phytoplankton development was noted on the border between the water column and bottom sediments. The phytoplankton biomass was uniquely high there (up to 130 g/m^3^) and was typical for hypertrophic water bodies [51]. Such development of phytoplankton sharply distinguished Lake Klyuchik from other water bodies, the Middle Volga basin, including the highly eutrophic Cheboksary reservoir [61], the Oka rivers [36], and others. 

Despite the higher alpha diversity (26 taxa) in the ecotone zone, compared to the other stations, monodominant phytoplankton communities were formed here, in which only one species of centric diatoms—*Cyclotella distinguenda*—had prevailed. A small part of the population of this diatom had a cell diameter of more than 20 microns [33]. It affected the proportion of large and small cell components in the algocenoses. The share of small cell algae in the biomass in this area of the lake was maximum. 

*Cyclotella*-species “blooming“ are often phenomena in the karstic lake, in both the temperate zone [16,20] and in the lakes of the warm belt [12,18,19], especially during spring and autumn turbulence. 

*Cyclotella distinguenda* is a rare species for the algoflora of the Volga basin, as well as for river systems in Hungary [62]. This species of centric diatoms, according to its ecological preferences, prefers mesotrophic conditions, and is sensitive to stratification [63]. In accordance with the literature data [16], similarly high values of biomass and abundance of this species were not observed in other lakes. We suppose that the unique combination and dynamics of the environmental factors of the lake (high mineralization, favorable light conditions, low temperature background, and lack of thermal stratification) were optimal for the mass development of this species and allowed it to regulate the structure and productivity of algocenoses in weak competition conditions. 

Such combinations of factors were likely not beneficial for the other groups of phytoplankton. Cyanobacteria are frequent components of phytoplankton communities in highly mineralized water bodies of the steppe and semi-desert zones [64,65,66], and in the subtropical zone [18,19], especially in the midsummer stratification period. In Lake Klyuchik, despite the high values of mineralization, the development and "blooming" of cyanobacteria were limited by low temperatures and by an absence in the western part of the reservoir, or weakly expressed stratification (in the eastern part). The cyanobacteria found in plankton were represented by small coccoid and filamentous forms (*Aphanocapsa* spp., *Pseudanabaena* spp.), but their abundance was not significant.

A similar trend can be noted for coccoid green algae, the diversity, growth rate, and productivity of which are stimulated by elevated temperatures [66,67].

The role of phytoflagellates (species from different taxonomic groups), frequent inhabitants of lentic water bodies [12,16,68], turned out to be less significant than other groups. Flagellar algae are weakly dependent on light conditions due to their greater tendencies to mixotrophic feeding and heterotrophic carbon assimilation [60,69,70], but they are sensitive to turbulence [19]. Since there was no pronounced stratification in the lake, accumulations of mobile monad algae at different stations of the lake could be formed in any part of the water column and their preference for a certain depth was not clearly manifested. However, the proportion of flagellate algae was statistically higher in the warm eastern part of the lake.

Among structural parameters, species richness did not show significant relations with other structural variables, whereas the number of dominant and subdominant species turned out to be more indicative and showed correlations. It means that the alpha diversity of phytoplankton (Shannon diversity index, Pielou evenness index and Simpson dominance index) were mainly determined by the number of the dominant species (not by the species richness of phytoplankton) and their ecological preferences (the ability to vegetate at low summer temperatures in mesohaline conditions). 

The size structure of the phytoplankton community, as an important indicator of water ecosystem eutrophication [47], reflected the predominance of nannoplankton components in phytoplankton communities of Lake Klyuchik. The proportion of small-celled organisms increased in communities with more pronounced dominance and decreased with an increase in species diversity and evenness.

The saprobity of water bodies within the limnosaprobic levels of their pollution with organic matter changes, in parallel with the levels of trophicity, can be considered as structural indicators of communities [46]. The variations in the average values of the saprobity indices in Lake Klyuchik was within the framework of oligo-β-mesosaprobic water pollution, with higher values in the ecotone zone of the lake. As the saprobity enhanced, we observed a tendency of simplification of phytoplankton communities (a decrease in diversity and evenness) and an increase in the role of the small-cell fraction.

## 5. Conclusions

The investigation of the gypsum karstic lake with the unique abiotic environmental factors made it possible to identify the most important features of the structure and dynamics of phytoplankton communities developing in such conditions. These patterns expand our understanding about the diversity of phytoplankton communities, in general, and provide clarifications about their organizations under certain environmental scenarios. The ideas will be useful for studying the biodiversity of phytoplankton communities for lakes with unusual—as well as typical—parameters, to assist in the planning (i.e., in the protection of unique landscapes or habitats) and in the assessments of the ecological statuses of these lakes.

## Figures and Tables

**Figure 1 microorganisms-10-00386-f001:**
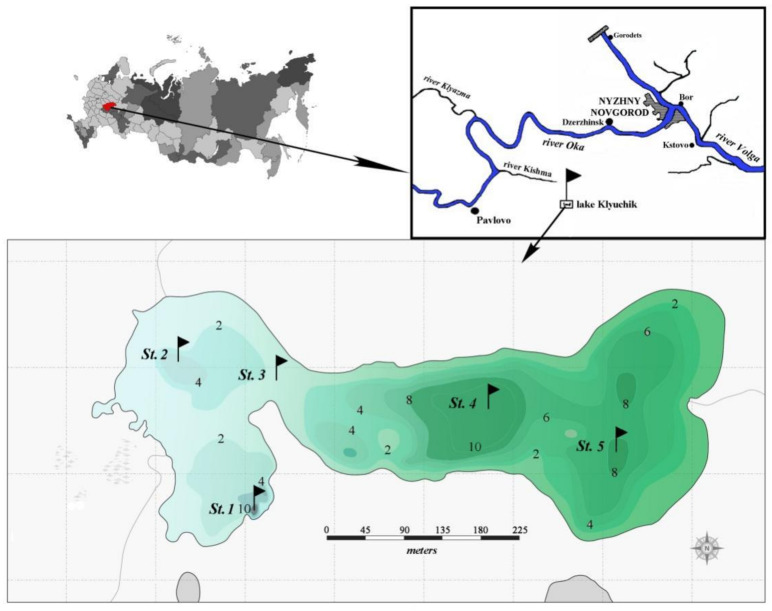
Map scheme of Lake Klyuchik. St. 1–St. 5—sampling stations; 2, 4, 6, 8, 10 are depths (meters) (according to www.lakemaps.org/ru, accessed on 20 November 2021, with some modifications).

**Figure 2 microorganisms-10-00386-f002:**
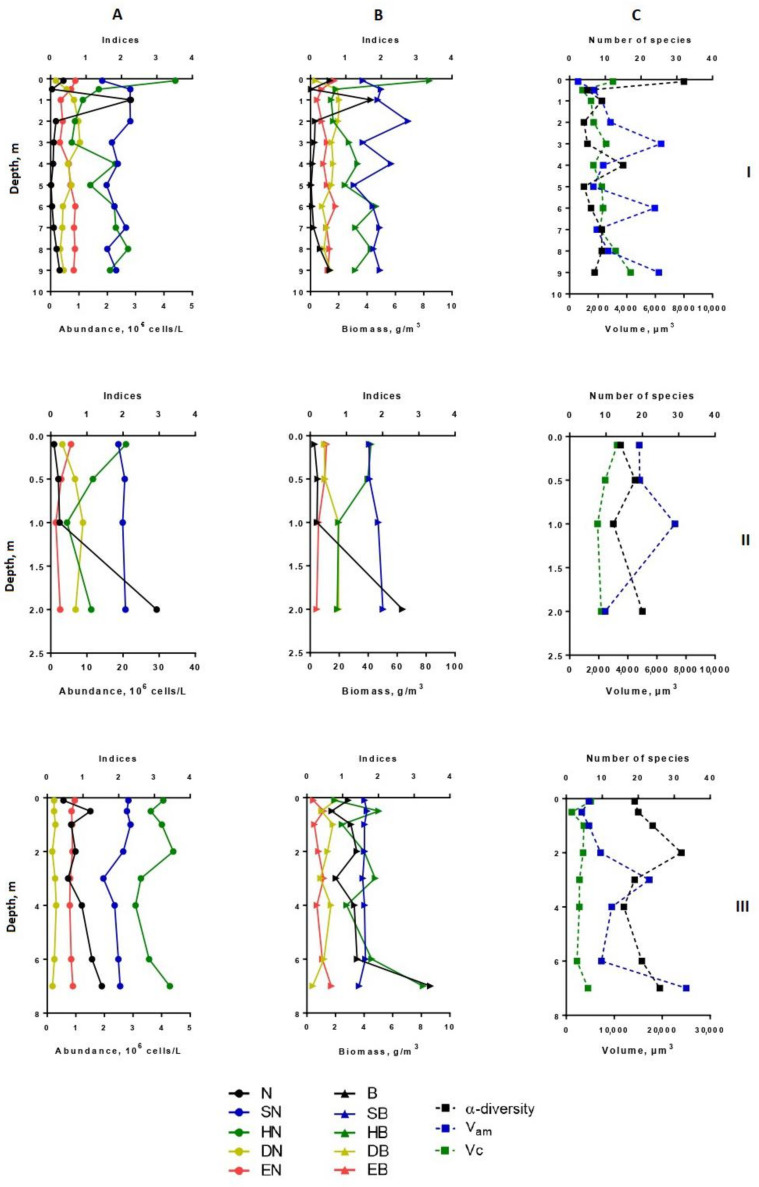
Vertical distribution of diversity indices and structural variables (α-diversity, abundance, biomass, saprobity, size structure) of phytoplankton in Lake Klyuchik in 2017 (July). A—Indicators for the abundance; B—indicators for biomass; C—the number of species and size structures, I—Station 1, II—Station 3, III—Station 5. N, 10^6^ cells/L—abundance; B, g/m^3^—biomass; SN–Pantle–Buck index, calculated to abundance; SB—Pantle–Buck index, calculated to biomass; HN bit/Ex—Shannon–Weaver diversity index for abundance; HB, bit/g—Shannon–Weaver diversity index for biomass; DN—Simpson dominance index for abundance; DB—Simpson dominance index for biomass; EN—Pielou evenness index for abundance; EB—Pielou evenness index for biomass; α-diversity—number of species per sample; V_am_—arithmetic volume; Vc, µm^3^—coenocytic volume.

**Figure 3 microorganisms-10-00386-f003:**
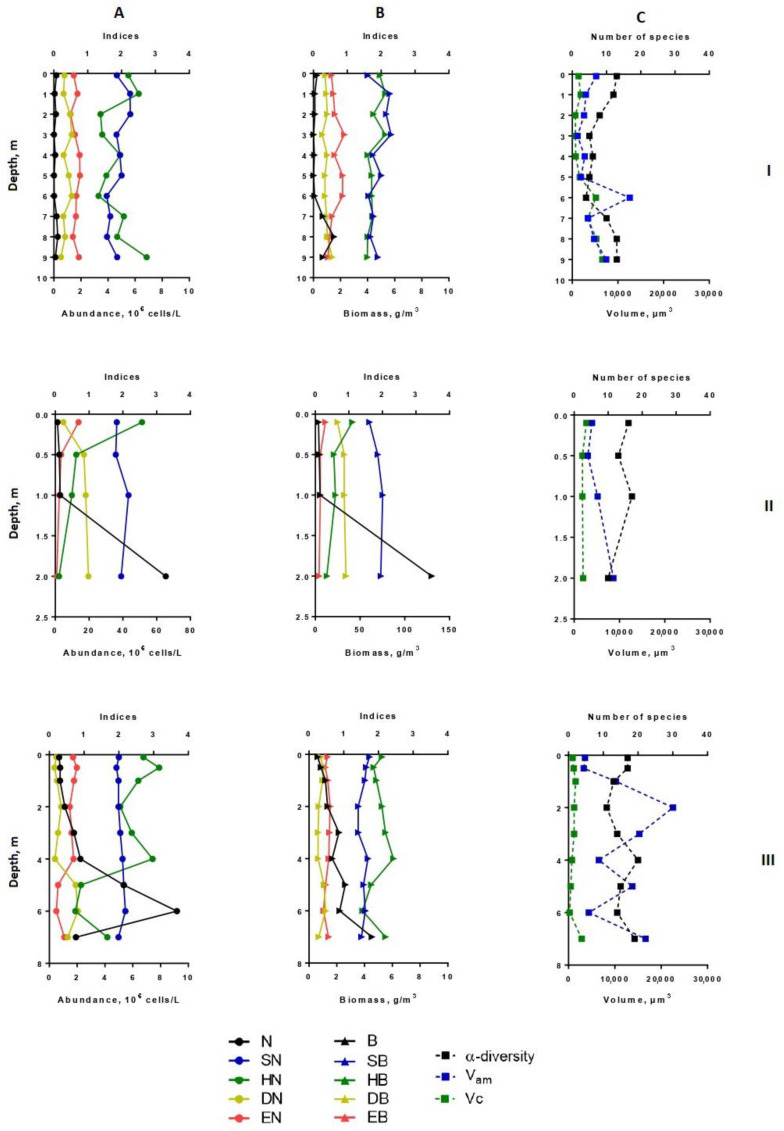
Vertical distribution of diversity indices and structural variables (α-diversity, abundance, biomass, saprobity, size structure) of phytoplankton in Lake Klyuchik in 2017 (August). A—Indicators for the abundance; B—indicators for biomass; C—the number of species and size structures, I—Station 1, II—Station 3, III—Station 5. N, 10^6^ cells/L—abundance; B, g/m^3^—biomass; SN—Pantle–Buck index, calculated to abundance; SB—Pantle–Buck index, calculated to biomass; HN bit/Ex—Shannon–Weaver diversity index for abundance; HB, bit/g—Shannon–Weaver diversity index for biomass; DN—Simpson dominance index for abundance; DB—Simpson dominance index for biomass; EN—Pielou evenness index for abundance; EB—Pielou evenness index for biomass; α-diversity—number of species per sample; V_am_—arithmetic volume; Vc, µm^3^—coenocytic volume.

**Figure 4 microorganisms-10-00386-f004:**
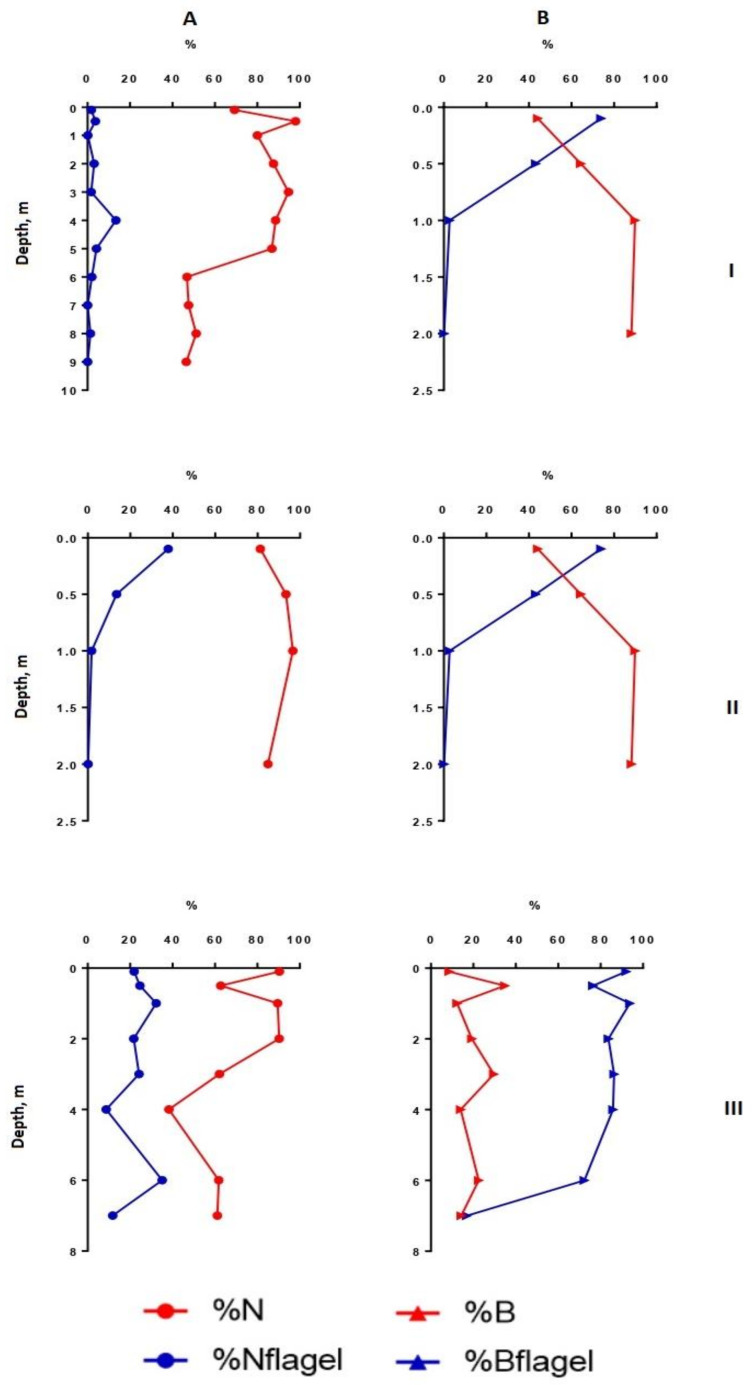
Vertical distribution of relative structural variables of phytoplankton in Lake Klyuchik in 2017 (July). A—Share of small-celled fraction and flagellar species proportion counted for the abundance, B—share of small-celled fraction and flagellar species proportion counted for the biomass; I—Station 1, II—Station 3, III—Station 5. %N—a share of a small cell fraction (<20 µm) in the total phytoplankton abundance; %B—a share of a small cell fraction (<20 µm) in the total phytoplankton biomass; %Nflagel—a share of the monad forms in total abundance of algae; %Bflagel—a share of the monad forms in total biomass of algae.

**Figure 5 microorganisms-10-00386-f005:**
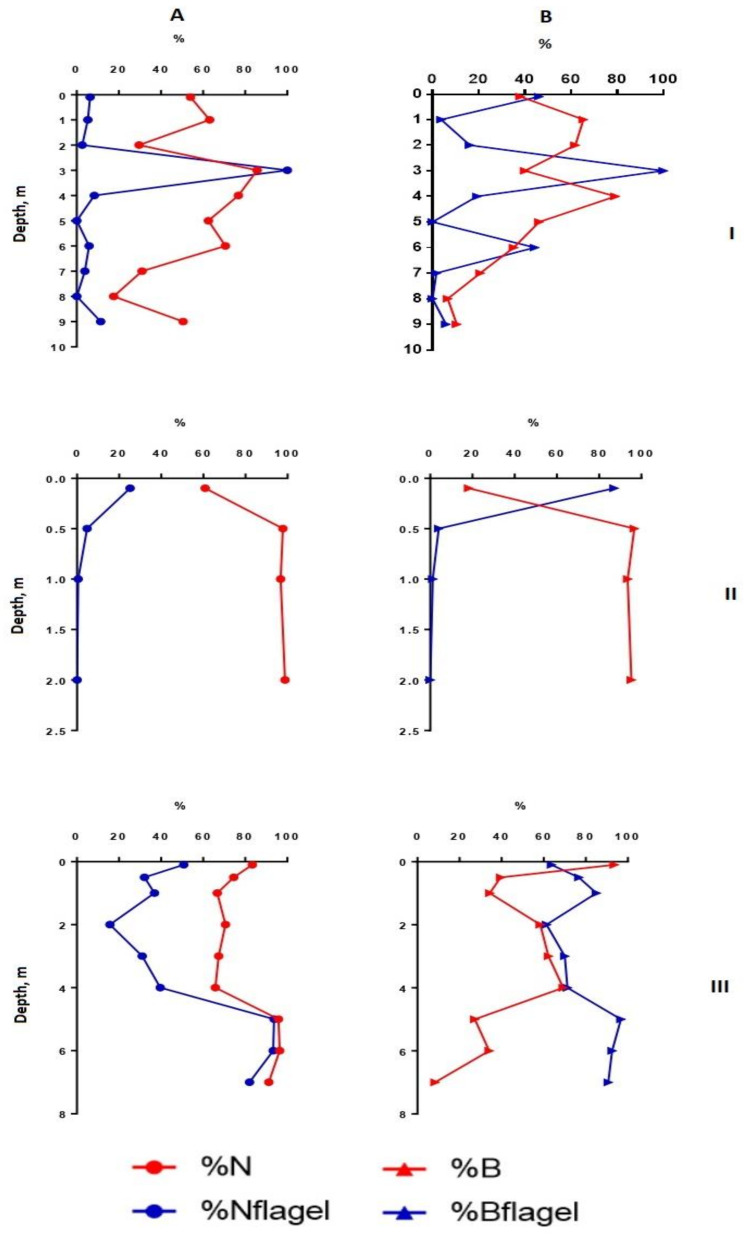
Vertical distribution of relative structural variables of phytoplankton in Lake Klyuchik in 2017 (August). A—Share of small-celled fraction and flagellar species proportion counted for the abundance, B—share of small-celled fraction and flagellar species proportion counted for the biomass; I—Station 1, II—Station 3, III—Station 5. %N – a share of a small cell fraction (<20 µm) in the total phytoplankton abundance; %B—a share of a small cell fraction (<20 µm) in the total phytoplankton biomass; %Nflagel—a share of the monad forms in total abundance of algae; %Bflagel—a share of the monad forms in total biomass of algae.

**Table 1 microorganisms-10-00386-t001:** Limnological, physical, and chemical variables in different stations of Lake Klyuchik, mean values with standard deviations, summer 2017.

Parameter	St. 1	St. 2	St. 3	St. 4	St. 5
Depth (m)	10.7	3.0	2.0	9.4	7.0
Transparency (m)	8.0–8.5	to the bottom	to the bottom	3.4–5.4	3.0–4.5

Temperature (°C)	7.5–9.5 8.8 ± 0.9	8.0–10.5 9.2 ± 0.7	8.1–13.6 10.6 ± 1.6	11.0– 15.8 13.8 ± 1.3	13.2–17.3 15.6±1.2

pH	7.0–7.1 7.1 ± 0.04	7.1–7.2 7.1 ± 0.03	7.1–7.6 7.3 ± 0.16	7.3–7.4 7.4 ± 0.02	7.4–7.6 7.5 ± 0.06

**Table 2 microorganisms-10-00386-t002:** Structural variables of phytoplankton in Lake Klyuchik, mean values with standard deviation, summer 2017.

Structural Variables	St. 1	St. 2	St. 3	St. 4	St. 5	U Criterion St. 1 × St. 5
N	0.31 ± 0.07	3.57 ± 2.10	25.80 ± 11.40	1.90 ± 0.41	1.53 ± 0.65	
B	0.94 ± 0.12	11.20 ± 6.00	55.50 ± 21.90	8.32 ± 0.48	6.82 ± 2.96	*p* ≤ 0.05
SN	1.59 ± 0.04	1.54 ± 0.02	1.94 ± 0.20	1.52 ± 0.01	1.64 ± 0.07	
SB	1.55 ± 0.04	1.75 ± 0.13	1.87 ± 0.05	1.62 ± 0.07	1.49 ± 0.08	
HN	2.25 ± 0.46	1.56 ± 0.21	1.01 ± 0.56	2.83 ± 0.02	2.22 ± 0.07	
HB	2.19 ± 0.42	1.61 ± 0.23	1.31 ± 0.70	3.31 ± 0.17	2.23 ± 0.18	
DN	0.37 ± 0.10	0.47 ± 0.04	0.71 ± 0.16	0.22 ± 0.01	0.34 ± 0.03	
DB	0.36 ± 0.11	0.50 ± 0.08	0.64 ± 0.20	0.14 ± 0.02	0.36 ± 0.06	
EN	0.58 ± 0.06	0.38 ± 0.04	0.21 ± 0.10	0.66 ± 0.01	0.51 ± 0.03	
EB	0.57 ± 0.09	0.40 ± 0.06	0.27 ± 0.13	0.77 ± 0.02	0.53 ± 0.07	
%N	74.00 ± 3.50	94.60 ± 1.60	91.60 ± 3.70	74.5 ± 2.60	73.40 ± 7.20	*p* ≤ 0.05
%B	59.80 ± 8.50	79.70 ± 9.40	82.80 ± 9.00	38.80 ± 5.10	44.60 ± 18.50	*p* ≤ 0.05
%Nflagel	0.60 ± 0.30	1.10 ± 0.20	0.20 ± 0.10	8.40 ± 5.90	11.30 ± 4.70	*p* ≤ 0.05
%Bflagel	0.40 ± 0.30	6.10 ± 3.70	1.00 ± 0.90	0.90 ± 0.70	14.30 ± 12.30	*p* ≤ 0.05
SDN	1.70 ± 0.30	2.00 ± 0.00	1.30 ± 0.30	3.00 ± 0.60	3.00 ± 0.60	
SDB	2.70 ± 0.30	2.30 ± 0.30	1.00 ± 0.00	2.70 ± 0.30	1.70 ± 0.30	
SMN	3.70 ± 0.30	2.30 ± 0.30	2.00 ± 0.60	3.30 ± 0.30	3.70 ± 0.30	
SMB	3.70 ± 0.30	3.70 ± 0.90	1.70 ± 0.30	6.70 ± 0.30	3.30 ± 0.70	
V_am_	4.60 ± 1.10	12.50 ± 4.10	7.80 ± 1.90	16.20 ± 4.90	19.90 ± 4.70	
Vc	3.30 ± 0.50	3.30 ± 0.20	2.30 ± 0.50	4.80 ± 0.90	5.60 ± 1.90	
Sp	18.30 ± 5.30	17.30 ± 1.80	26.00 ± 4.60	20.00 ± 1.70	21.00 ± 3.20	

**Table 3 microorganisms-10-00386-t003:** Spearman correlation matrix of the diversity indices of phytoplankton in Lake Klyuchik (*p* ≤ 0.05).

	HB	HN	DB	DN	EB	EN
HB	1.00	0.88	−0.99	−0.89	0.86	0.88
HN	0.88	1.00	−0.87	−0.96	0.74	0.95
DB	−0.99	−0.87	1.00	0.87	−0.88	−0.85
DN	−0.89	−0.96	0.87	1.00	−0.73	−0.91
EB	0.86	0.74	−0.88	−0.73	1.00	0.78
EN	0.85	0.95	−0.85	−0.91	0.78	1.00

**Table 4 microorganisms-10-00386-t004:** The correlation coefficients found by correlation analysis of α-diversity indices and some structural variables of phytoplankton communities in Lake Klyuchik (*p* ≤ 0.05).

	HB	HN	DB	DN	EB	EN
N	*p* ≥ 0.05	*p* ≥ 0.05	*p* ≥ 0.05	*p* ≥ 0.05	*p* ≥ 0.05	−0.54
SB	*p* ≥ 0.05	−0.52	*p* ≥ 0.05	*p* ≥ 0.05	*p* ≥ 0.05	−0.55
%N	−0.62	−0.76	0.59	0.76	−0.66	−0.82
%B	−0.71	−0.85	0.73	0.74	−0.67	−0.81
%Nflagel	0.57	0.64	−0.54	−0.77	*p* ≥ 0.05	0.61
SDN	0.70	0.64	−0.65	−0.78	0.56	0.57
SMN	0.56	0.77	*p* ≥ 0.05	−0.75	*p* ≥ 0.05	0.74
SDB	*p* *≥* *0.05*	*p* ≥ 0.05	*p* ≥ 0.05	*p* ≥ 0.05	*p* ≥ 0.05	0.53
SMB	0.74	0.67	−0.77	−0.74	0.68	0.76

## Data Availability

No new data were created or analyzed in this study. Data sharing is not applicable to this article.

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
