# Peer review of "Phytoplankton Community Structure in Highly-Mineralized Small Gypsum Karst Lake (Russia)"

_microorganisms, 2022, doi:10.3390/microorganisms10020386_

Round 1
Reviewer 1 Report
Principal advantage of the study is the detailed description of the lake and careful data analysis. It is good to have complex environmental and phytoplankton data, which can be used in the future monitoring of the lake.
However, the structure of the ms is really not good. Scientific papers should be written according to the strict rules, because this allows the readers to understand them faster and to find specific information at its specific place. The “Results” should contain only the facts, which were obtained during the research. In particular, this means that no more than 1-2 citations are acceptable in this chapter. I counted at least 9 citations, which authors put to their “Results”. All discussion should be in the “Discussion” chapter. No observations of the facts should be in the Discussion, only comparisons and authors thoughts. Thus, usually, Discussion is bigger than Results… Conclusions should include only one small paragraph, where the authors summarize the main point of their paper. It should not be as an abstract and should not include numbers.
English language of the ms should be improved as well.
Authors cited many of their own papers (it is understandable, because nobody else study their lake) and papers concerning general limnology. There are too few citations of other studies of similar (or opposite...) water reservoirs (preferable to find such citations in English and with doi). Could you place your study to the broader context (not just Volga basin)? I do believe, it is possible. What can we study from the such model as your lake?
How phytoplankton changed from June to August in different stations, were the changes typical? (perhaps, it was written in the Results, but I did not get it. It should be discussed in the Discussion)
Do you have any data concerning the studied lake from previous years, which can be taken for the comparison with your current research?
Reviewer 2 Report
The manuscript of Okhapkin et al. refers to the description of the phytoplankton community in lake Klyuchik (Russia). Although the manuscript carries relevant data it lacks some insights and clarifications. To begin what is the purpose of the study? I mean why was this lake chosen and why were taxonomic methods employed? Why what is the distinction of this study to others with metagenomics data? Another issue is why were no toxicity assays conducted? Is the lake being monitored and why the need for a taxonomic composition in this region? Is the lake sensible to eutrophication? Is it surrounded by pollution sources? What is the N and P and are these being monitored in the lake?
Regarding the data retrieved from the study why were cyanobacteria poorly dominated if the monitoring occurred between June and August? Is there any episodes of blooms in the lake? Why is no toxicity data available?
Finally what are the authors contribution to the study of taxonomic composition in a given ecosystem? I mean at present why not a metagenomics analysis and only a morphologic study? And if these type of studies are considered what is their main advantage to the study of the phytoplankton community and its surveillance in other ecosystems worldwide?
Round 2
Reviewer 1 Report
The new version of the ms looks much better. The data is well analyzed and put in a broad context. Authors used "old fashioned" method (optical microscopy), but I agree with them, that "modern" methods (like DNA metabarcoding) will in general show similar patterns. However, I encourage them to use also molecular genetic approach: it may allow to find interesting rare taxa, to study phylogeography (and nowadays it is enough data from other lakes for the comparison).
I think, it is better to write some conclusion at the end: 2-3 sentences about the the main advantages of their work and how this content will be used in future. I also suggest authors to recheck their text (typos exist).
Author Response
Dear Reviewer,
Thank you for helpful recommendations.
We have added Conclusion in the new version of manuscript. We also tried to rechecked carefully the text for typos.
Reviewer 2 Report
The authors have answered to my comments and now the manuscript warrants publication in Microorganisms.
Author Response
Dear Reviewer,
Thank you for reviewing our manuscript and offering helpful reccomendations.